MIT-CTP/5802

# A Lorentz-Equivariant Transformer for All of the LHC

Johann Brehmer[1], Víctor Bresó[2], Pim de Haan[1], Tilman Plehn[2,3],
Huilin Qu[4], Jonas Spinner[2], and Jesse Thaler[5,6]

**1** CuspAI, Amsterdam, the Netherlands
**2** Institute for Theoretical Physics, Universität Heidelberg, Germany
**3** Interdisciplinary Center for Scientific Computing (IWR), Universität Heidelberg, Germany
**4** CERN, EP Department, Geneva, Switzerland
**5** Center for Theoretical Physics, Massachusetts Institute of Technology, Cambridge, MA, USA
**6** The NSF AI Institute for Artificial Intelligence and Fundamental Interactions

December 24, 2024

## Abstract

We show that the Lorentz-Equivariant Geometric Algebra Transformer (L-GATr) yields state-of-the-art performance for a wide range of machine learning tasks at the Large Hadron Collider. L-GATr represents data in a geometric algebra over space-time and is equivariant under Lorentz transformations. The underlying architecture is a versatile and scalable transformer, which is able to break symmetries if needed. We demonstrate the power of L-GATr for amplitude regression and jet classification, and then benchmark it as the first Lorentz-equivariant generative network. For all three LHC tasks, we find significant improvements over previous architectures.

# 1 Introduction

Modern machine learning (ML) is poised to define much of the future program at the Large Hadron Collider (LHC), from triggering and data acquisition to object identification, anomaly searches, non-perturbative input, theory and detector simulations, and simulation-based inference. The question is no longer if deep neural networks will be used for all of these purposes; it is how we can ensure that these networks provide optimal and resilient results, including a comprehensive uncertainty treatment [1].

Early ML studies for the LHC assumed that there would be no shortage of training data, because precision simulations were close enough to the actual LHC data to train neural networks for a diverse range of purposes. These studies assumed the availability of labeled and fully understood simulated samples, such that trained networks could be reliably applied to the limited actual LHC data. Now, however, this point of view is starting to get challenged, because:

1. standard architectures do not capture amplitudes or densities at the per-mille level;
2. the target precision for LHC applications requires huge datasets even by cheap simulation standards; and
3. small deviations between simulated and measured data require networks to be tuned on measurements.

One way to tackle these challenges is to use our knowledge about the structure of LHC data. In particle physics, much of this knowledge is reflected in complex symmetry structures, from the detector geometry to the relativistic phase space, eventually including the underlying theory. Learning the Minkowski metric is a known challenge for all networks working on relativistic phase space, so an obvious question is if one could work with Lorentz-invariant or Lorentz-covariant internal representations and save the networks the time and effort to learn such a representation for each task.

The first modern ML applications in LHC physics were jet taggers [2, 3], which aimed to optimally analyze the substructure of jets to identify their partonic nature [4, 5] based on all available low-level information [4–8]. These jet taggers also triggered the question of how and whether to include Lorentz symmetry [9], leading to a set of Lorentz-equivariant jet taggers, currently under experimental study [10–14].* Targeting the ultimate tagging precision with limited training data, these equivariant taggers paved the way for a new phase of precision-machine learning for the LHC, where established ML methods have to be systematically enhanced for and by particle physics.

Any inference from LHC data is reliant on precision simulations of events and detectors through a combination of first-principles theory and physics-motivated modeling [15, 16], making LHC physics a prime target for the generative AI revolution [17]. On the detector side, simple generative adversarial networks (GANs) as part of the fast ATLAS detector simulation [18] have been generalized to generic detectors [19, 20] with a wide range of networks targeting the combination of high multiplicity, sparsity, and accuracy [21]. On the theory side, this program has developed from the first ML-generators for partonic LHC events [22–24] into a comprehensive generative-network program in event generators [25–32].

Our goal is to provide a way to systematically exploit Lorentz equivariance in different networks tasks defined on relativistic phase space. A Lorentz-equivariant transformer can provide the appropriate internal or latent representation of phase space data to regression, classification, and generative networks. L-GATr is based on the Geometric Algebra Transformer (GATr) [33, 34], designed for Euclidean translations, rotations, and reflections. We generalize

---

*In physics, we typically use the term "covariant" rather than "equivariant", but equivariant networks are an established subfield of ML, so we stick to this term.

it to L-GATr encoding Lorentz-equivariance into new network layers, including a maximally expressive linear map, attention, and layer normalization. This architecture was originally developed for an ML audience and applied to amplitude regression, top tagging, and event generation in Ref. [35]. In this physics-targeted study, we extend the amplitude regression part, improve the classification through pre-training and multi-class tagging, and deliver a competitive generative network. We also provide a comprehensive benchmarking with the state of the art for these three distinct LHC applications.

After reviewing the construction of L-GATr in Sec. 2, we demonstrate the versatility of L-GATr through three LHC case studies. In Sec. 3 we show how it allows us to efficiently learn precision surrogates for scattering amplitudes up to a $Z + 5$ gluon final state. Next, we show in Sec. 4 how we can improve transformer-based jet taggers with an equivariant setup. We show how L-GATr benefits from pre-training for the ultimate performance and can be generalized to multi-class tagging. Finally, in Sec. 5 we employ L-GATr inside a diffusion generator and show how it generates LHC events for instance for final states up to $t\bar{t} + 4$ jets better than all benchmarks. We provide a brief summary and outlook in Sec. 6. Throughout the text and for all figures we indicate possible overlap with Ref. [35].

## 2 Lorentz-Equivariant Geometric Algebra Transformer

In this section, we discuss the principles behind L-GATr and its most important features. L-GATr uses the geometric algebra, a mathematical framework that represents certain geometric objects and operations in a unified language. Using the language of geometric algebra, it is straightforward to build equivariant layers while retaining most of the structure from typical neural networks. In cases where the Lorentz symmetry is not completely realized, we show how reference inputs can be used to make L-GATr equivariant with respect to specific subgroups of the Lorentz group. Finally, we study how L-GATr scales with the phase space dimensionality, as compared to other network architectures.

### 2.1 Spacetime Geometric Algebra

A geometric algebra is defined as an extension of a vector space with an extra composition law — the geometric product [36]. The geometric product of two vectors $x$ and $y$ is decomposed into a symmetric and an antisymmetric contribution,

$$xy = \frac{\{x, y\}}{2} + \frac{[x, y]}{2} \,, \tag{1}$$

where the anti-commutator $\{x, y\}/2$ represents the usual inner product, and $[x, y]/2$ constitutes a new outer product. This second term defines the bivector, which can be understood geometrically as an area element of the plane spanned by $x$ and $y$. The geometric product $xy$ is an element of the algebra made up by the sum of a scalar and a bivector. Neither object is part of the original vector space, so the geometric product extends the original vector space. This extension can be carried out systematically by applying the geometric product repeatedly to the basis elements of the vector space. To develop L-GATr, we focus on the etric alg $\mathbb{G}_{1,3}$, built from the vector space $\mathbb{R}^4$ with the metric $g = \text{diag}(1, -1, -1, -1)$. We choose as a basis for the vector space a set of four real vectors $\gamma^\mu$, which satisfy the anti-commutation relation

$$\{\gamma^\mu, \gamma^\nu\} = 2g^{\mu\nu} \,. \tag{2}$$

This inner product establishes the basis elements as a set of orthogonal vectors and fixes their normalization. This happens to also be the defining property of the gamma matrices, the basis

elements of the Dirac algebra used to describe spinor interactions. Both algebras are closely related, the only difference being that the spacetime algebra is defined over $\mathbb{R}^4$, whereas the Dirac algebra is defined over $\mathbb{C}^4$. This prescription fully recovers all the algebra properties presented in Ref. [35].

We now construct new elements of the algebra using the geometric product defined in Eq. (1). All higher-order elements can be characterized as antisymmetric products of $\gamma^\mu$. We organize them in grades, defined by the number of $\gamma^\mu$ needed to express them. For instance, the antisymmetric tensor $\sigma^{\mu\nu}$ is generated from the geometric product of two $\gamma^\mu$ and consequently has grade two,

$$\gamma^\mu \gamma^\nu = \frac{\{\gamma^\mu, \gamma^\nu\}}{2} + \frac{[\gamma^\mu, \gamma^\nu]}{2} = g^{\mu\nu} + \sigma^{\mu\nu} \,. \tag{3}$$

Following Eq. (1), $\sigma^{\mu\nu}$ is a bivector, which can be interpreted as the plane opened by $\mu$ and $\nu$ in Minkowski space. We see that the symmetric term in the geometric product reduces the grade, while the antisymmetric term increases it. The whole product $\gamma^\mu \gamma^\nu$ is a sum of grade zero (scalar) and grade two. A generic element of the algebra that mixes grade information is called a multivector.

Moving on, the geometric product of three vectors $\gamma^\mu \gamma^\nu \gamma^\rho$ contains the antisymmetric tensor $\epsilon_{\mu\nu\rho\sigma} \gamma^\mu \gamma^\nu \gamma^\rho$ as a trivector or axial vector. The product of all four $\gamma^\mu$ leads us to the pseudoscalar

$$\gamma^5 = \gamma^0 \gamma^1 \gamma^2 \gamma^3 \equiv \frac{1}{4!} \epsilon_{\mu\nu\rho\sigma} \gamma^\mu \gamma^\nu \gamma^\rho \gamma^\sigma. \tag{4}$$

Pseudoscalars act as parity reversal operations on any object and can be used to write axial vectors as $\gamma^\mu \gamma^5$. The missing factor $i$ compared to the usual definition of $\gamma^5$ indicates the slight difference between the complex Dirac algebra and our real spacetime algebra.

Geometric products with more than four $\gamma^\mu$ can be reduced to lower-grade structures. Combining all these elements, we can express any multivector of the algebra as[†]

$$x = x^S \, 1 + x^V_\mu \, \gamma^\mu + x^B_{\mu\nu} \, \sigma^{\mu\nu} + x^A_\mu \, \gamma^\mu \gamma^5 + x^P \, \gamma^5 \qquad \text{with} \qquad \begin{pmatrix} x^S \\ x^V_\mu \\ x^B_{\mu\nu} \\ x^A_\mu \\ x^P \end{pmatrix} \in \mathbb{R}^{16} \,. \tag{5}$$

In this representation, we only include the nonzero and independent entries in the antisymmetric bivector. Multivectors can be used to represent both spacetime objects and Lorentz transformations. For instance, particles are characterized by their type (i.e. particle identification, or PID) and their 4-momentum $p^\mu$,

$$x^S = \text{PID} \qquad x^V_\mu = p_\mu \qquad x^T_{\mu\nu} = x^A_\mu = x^P = 0 \,. \tag{6}$$

Using this convenient representation, the spacetime algebra naturally structures relevant objects like parity-violating transition amplitudes. The matrix element $\mathcal{M}$ is a function of 4-momenta and can be decomposed into parity-even and parity-odd terms before it gets squared,

$$|\mathcal{M}|^2 = |\mathcal{M}_E|^2 + |\mathcal{M}_O|^2 + 2\,\text{Re}\left(\mathcal{M}_E^* \mathcal{M}_O\right) \,. \tag{7}$$

---

[†]Some of us are reminded of supersymmetric multiplets, which also combine fields with different transformation properties into a graded structure. In that case, the elements of the multiplets are defined by a closing under supersymmetry transformations with spinor-like generators and an (anti-)commutator-defined algebra. In superspace, the elements of the multiplets can be extracted using a finite expansion in Grassmann variables. One difference between superfields and our multivectors is that in supersymmetry, it is known how to incorporate all irreducible representations relevant for phenomenology (e.g. vector and chiral superfields), while for our multivectors, it is not obvious how to extend the space to higher-rank representations.

The first two terms represent a scalar function of the 4-momenta, while the last is a pseudoscalar function. To calculate this amplitude in the spacetime algebra, we first embed each 4-momentum into a multivector $x^i = p^i_\mu \gamma^\mu$. Using these multivectors as inputs, the squared amplitude can be obtained through a sequence of algebra operations. The result of this calculation will also be a multivector, namely

$$|\mathcal{M}|^2 = x^S 1 + x^P \gamma^5 \qquad \text{with} \qquad \begin{aligned} x^S 1 &= |\mathcal{M}_E|^2 + |\mathcal{M}_O|^2 \\ x^P \gamma^5 &= 2\,\text{Re}\left(\mathcal{M}_E^* \mathcal{M}_O\right). \end{aligned} \tag{8}$$

The geometric algebra explicitly separates the scalar and pseudoscalar components of the squared amplitude, highlighting their respective geometric significance.

The geometric algebra also allows us to perform operations on spacetime objects. Lorentz transformations act as

$$\Lambda_v(x) = v x v^{-1}, \tag{9}$$

where $v$ is a multivector representing an element of the Lorentz group acting on the algebra element $x$, and $v^{-1}$ representings the corresponding inverse. The representation $v$ of a Lorentz transformation is built by a simple rule: a multivector encoding an object that is invariant under a Lorentz transformation will also represent the transformation itself. This gives a dual interpretation to spacetime algebra elements as, both, geometric objects and Lorentz transformations.

For instance, boosts along the $z$-axis are generated by $\sigma^{03}$, which also represents a plane in time vs. $z$-direction. The multivector for such a boost with rapidity $\omega$ reads

$$v = e^{\omega\sigma^{03}} = 1\cosh\omega + \sigma^{03}\sinh\omega. \tag{10}$$

If we apply this boost to a particle moving in $z$-direction, $x = E\gamma^0 + p_z\gamma^3$, the transformation in Eq. (9) gives us

$$v x v^{-1} = (E\cosh\omega - p_z\sinh\omega)\gamma^0 + (p_z\cosh\omega - E\sinh\omega)\gamma^3. \tag{11}$$

This is exactly what we expect from the Lorentz boost. The algebra representation allows us to apply this boost on any object in the geometric algebra, not just vectors. From Eq. (9) and the properties of the geometric product, we see that Lorentz transformations will never mix grades. Each algebra grade transforms under a separate sub-representation of the Lorentz group.

The main limitation of the geometric algebra approach is that the spacetime algebra $\mathbb{G}_{1,3}$ covers only a limited range of Lorentz tensor representations. For instance, this formalism can not represent symmetric rank-2 tensors. For most LHC applications, though, one does not encounter higher-order tensor representations as inputs or outputs, so this is not a substantial limitation. Whether higher-order tensors might be needed for internal representations within a network is an open question [37].

## 2.2 Constructing a Lorentz-Equivariant Architecture

Based on the multivector representation, we now construct the corresponding transformer network L-GATr. It is equivariant under Lorentz group transformations $\Lambda$

$$\text{L-GATr}\left(\Lambda(x)\right) = \Lambda\left(\text{L-GATr}(x)\right). \tag{12}$$

We take advantage of the fact that multivector grades form sub-representations of the Lorentz group, i.e. all multivector components of the same grade transform equally under all network operations, whereas different grades transform differently [33, 35].

The L-GATr architecture uses variations of the standard transformer operations Linear, Attention, LayerNorm,tion, adapted to process multivectors [38, 39]. As usual for transformers, the input $x$ and output L-GATr$(x)$ are unordered sets of $n_t$ ordered lists of $n_c$ multivector channels

$$
x_{ic} = \begin{pmatrix} x_{ic}^S \\ x_{\mu,ic}^V \\ x_{\mu\nu,ic}^B \\ x_{\mu,ic}^A \\ x_{ic}^P \end{pmatrix} \qquad i = 1, ..., n_t \quad c = 1, ..., n_c \, . \tag{13}
$$

We call the set elements $x_i = \{x_{ic} : c = 1, ..., n_c\}$ tokens, where each token can represent a particle. In the network, every operation will have multivectors as inputs and outputs. The full L-GATr architecture is built as

$$
\bar{x} = \text{LayerNorm}(x)
$$
$$
\text{AttentionBlock}(x) = \text{Linear} \circ \text{Attention}(\text{Linear}(\bar{x}), \text{Linear}(\bar{x}), \text{Linear}(\bar{x})) + x
$$
$$
\text{MLPBlock}(x) = \text{Linear} \circ \text{Activation} \circ \text{Linear} \circ \text{GP}(\text{Linear}(\bar{x}), \text{Linear}(\bar{x})) + x
$$
$$
\text{Block}(x) = \text{MLPBlock} \circ \text{AttentionBlock}(x)
$$
$$
\text{L-GATr}(x) = \text{Linear} \circ \text{Block} \circ \text{Block} \circ \cdots \circ \text{Block} \circ \text{Linear}(x) \, . \tag{14}
$$

We define the modified transformer operations in some detail:

- For the linear layers, we use the fact that equivariant operations on multivectors process components within the same grade equally. We use the projection $\langle \cdot \rangle_k$ to extract the $k$-grade and apply different learnable coefficients for each grade. As a result, the most general linear combination of independently-transforming multivector components is

$$
\text{Linear}(x) = \sum_{k=0}^{4} v_k \langle x \rangle_k \quad \left( + \sum_{k=0}^{4} w_k \gamma^5 \langle x \rangle_k \right) , \tag{15}
$$

  where $v, w \in \mathbb{R}^5$ are learnable parameters and $k$ runs over the five algebra grades. The second term is optional and breaks the symmetry down to the special orthochronous Lorentz group, the fully-connected subgroup that leaves out parity and time reversal. In this subgroup, discrete transformations are not present, so any pair of algebra elements that differ by a $\gamma^5$ factor can be linearly mixed without breaking equivariance.

- We extend scaled dot-product attention such that it can be applied to multivectors

$$
\text{Attention}(q, k, v)_{ic} = \sum_{j=1}^{n_t} \text{Softmax}_j \left( \sum_{c'=1}^{n_c} \frac{\langle q_{ic'}, k_{jc'} \rangle}{\sqrt{16 n_c}} \right) v_{jc} \, , \tag{16}
$$

  where $n_c$ is the number of multivector channels and $\langle \cdot, \cdot \rangle$ is the $\mathbb{G}_{1,3}$ inner product. This inner product can be pre-computed as a list of signs and a Euclidean inner product, allowing us to use standard transformer implementations.

- Layer normalization on multivectors is non-trivial because the $\mathbb{G}_{1,3}$ norm can have zero and negative contributions. For this reason, we define layer normalization using the absolute value of the inner product for each grade separately

$$
\text{LayerNorm}(x) = \frac{x}{\sqrt{\frac{1}{n_c} \sum_{c=1}^{n_c} \sum_{k=0}^{4} \left| \left\langle \langle x_c \rangle_k, \langle x_c \rangle_k \right\rangle \right| + \epsilon}} \, , \tag{17}
$$

  where $\epsilon$ is a normalization constant and $n_c$ is the number of multivector channels.

| Layer type | Transformer | L-GATr |
|---|---|---|
| Linear($x$) | $vx + w$ | $\sum_{k=0}^{4} v_k \langle x \rangle_k$ |
| Attention($q, k, v)_{ic}$ | $\sum_{j=1}^{n_t} \text{Softmax}_j \left( \sum_{c'=1}^{n_c} \frac{q_{ic'} k_{jc'}}{\sqrt{n_c}} \right) v_{jc}$ | $\sum_{j=1}^{n_t} \text{Softmax}_j \left( \sum_{c'=1}^{n_c} \frac{\langle q_{ic'}, k_{jc'} \rangle}{\sqrt{16 n_c}} \right) v_{jc}$ |
| LayerNorm($x$) | $x \left[ \frac{1}{n_c} \sum_{c=1}^{n_c} x_c^2 + \epsilon \right]^{-1/2}$ | $x \left[ \frac{1}{n_c} \sum_{c=1}^{n_c} \sum_{k=0}^{4} \left| \left\langle \langle x_c \rangle_k, \langle x_c \rangle_k \right\rangle \right| + \epsilon \right]^{-1/2}$ |
| Activation($x$) | GELU($x$) | GELU($\langle x \rangle_0)x$ |
| GP($x, y$) | — | $xy$ |

Table 1: Comparison of transformer layers and L-GATr layers. The arguments $x, y, q, k, v$ are scalars for the transformer, and multivectors for L-GATr. The second term in the L-GATr linear layer is optional and breaks the Lorentz group down to its fully connected subgroup.

- Activation functions applied directly on the multivectors break the equivariance. We employ scalar-gated activation functions [33], where the nonlinearity only acts on the scalar component of the multivector $\langle x \rangle_0$. Specifically, we use the scalar-gated GELU [40] activation function

$$\text{Activation}(x) = \text{GELU}(\langle x \rangle_0) \, x \, . \tag{18}$$

- Finally, the geometric algebra allows for another source of nonlinearity, the geometric product

$$\text{GP}(x, y) = xy \qquad \text{with} \qquad \text{GP}(vxv^{-1}, vyv^{-1}) = v\text{GP}(x, y)v^{-1} \, , \tag{19}$$

which is equivariant itself.

These operations strictly generalize standard scalar transformers to the multivector representation, as illustrated in Table 1. We supplement the list of multivector channels with extra scalar channels to allow a smooth transition to standard transformers that solely rely on scalar channels. Moreover, it provides a handle to feed large amounts of scalar information to the network without overloading the multivector channels.

## 2.3 Breaking Lorentz Symmetry

In many LHC contexts, Lorentz symmetry is only partially preserved. L-GATr can apply partial symmetry breaking in a tunable manner by including reference multivectors as additional inputs. Any network operation that involves such a reference vector will violate equivariance, breaking the symmetry group to a subgroup where the reference direction is fixed. This defines a partial symmetry breaking without altering the structure of the network. The network always has the option to tune out the reference vectors when they are not needed. Reference vectors are appended for each L-GATr input $x$ in the same way, either as extra tokens, or as extra channels within each token.

For instance, the LHC beam direction breaks the Lorentz group to the subgroup of rotations around and boosts along the beam axis [10, 14, 41, 42]. The natural reference vector is this beam direction, which can be either implemented as two vectors $x_{\pm}^V = (0, 0, 0, \pm 1)$, or one bivector representing the $x - y$ plane, $x_{12}^B = 1$. We find similar performance for both choices. Generally, we can break the Lorentz group to the subgroup of rotations in three-dimensional space $SO(3)$ using the reference multivector $x^V = (1, 0, 0, 0)$.

We include such reference multivectors as extra tokens for jet tagging in Sec. 4, and as extra channels for generation in Sec 5. In both cases, this symmetry breaking is crucial, and the specific way it is implemented has a strong impact on the network performance.

We find it beneficial to add more ways of breaking the symmetry that are formally equivalent to the reference multivectors discussed above. For jet tagging, we include additional kinematic inputs like $p_T, E, \Delta R$ embedded as scalars. These variables are only invariant under the subgroup of rotations around the beam axis, and L-GATr can reconstruct them based on the particles and reference multivectors. For event generation, we extract the $m$ and $p_T$ CFM-velocity components from scalar output channels of L-GATr and use them to overwrite the equivariantly predicted velocity components. We explain these aspects further in Sec. 4, 5.

## 2.4 Scaling with the Number of Particles

Fully connected graph neural networks bear a very close resemblance to transformers [43]; both process data as sets of tokens, both respect full or partial permutation symmetry, and both can be turned equivariant [10, 12, 14, 35]. Resource efficiency is where the two architectures differ most. To quantify it, we measure the scaling of speed and memory consumption of a standard transformer, L-GATr, and the Clifford Group Equivariant Neural Network (CGENN) [14], which is a graph network built on geometric algebra representations. We expect the CGENN to represent the strengths and limitations of equivariant graph networks. We execute network forward passes with synthetic data on an H100 GPU. Each measurement is done with a single event consisting of a varying number of particles. To ensure fairness, all networks consist of a single network block, for the transformers the attention receives inputs with 72 channels from 4 attention heads, and the CGENN is set up to be as close as possible to L-GATr. Namely, both contain 8 multivector channels and 16 scalar channels.

In the left panel of Fig. 1, we see that the evaluation time of all networks is independent of the number of tokens in the few-token regime, but it eventually scales quadratically. This transition happens when attention or message passing, rather than other parallelizable operations, becomes the limiting factor. L-GATr scales like a standard transformer in the many-token regime because they both use the same attention module. For few tokens, L-GATr is slower because of the more expensive linear layers. The CGENN is slower than L-GATr for few tokens,

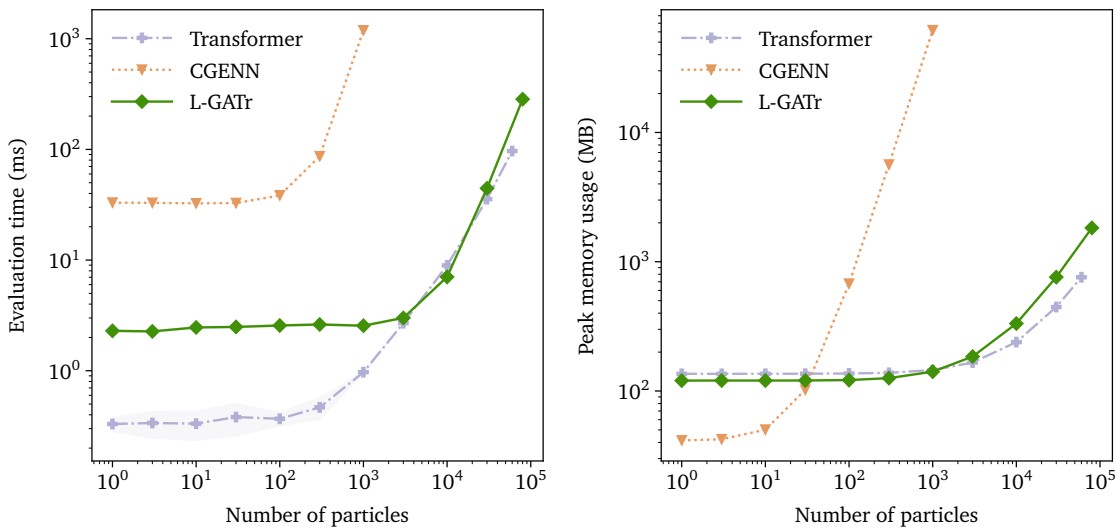

Figure 1: Scaling behavior of L-GATr, a standard transformer, and the equivariant graph network CGENN. The left panel was already discussed in Ref. [35].

and the quadratic scaling due to the expensive message passing operation takes off sooner.

As can be seen in the right panel of Fig. 1, for many particles L-GATr and the standard transformer display the same linear scaling in memory usage with the number of tokens since they use the same attention module. In contrast, the CGENN scales quadratically in this regime and runs out of memory already for 1000 particles. We attribute this to the different degree of optimization in the architectures. For L-GATr we use FlashAttention [44], heavily optimized for speed and memory efficiency. Graph neural networks are often optimized for sparsely connected graphs, so the efficiency of the standard implementation degrades for fully connected graphs.

## 3 L-GATr for Amplitude Regression

Our first L-GATr case study is for amplitude regression. Partonic scattering amplitudes can be calculated exactly as a function of phase space. However, their evaluation can become very expensive if we include high-order corrections and a large number of external particles. Amplitude surrogates as part of standard event generators speed up these precision predictions [45–48]. However, standard neural networks struggle to reach sufficient accuracy for realistic numbers of external particles. L-GATr uses the partial permutation symmetry of particles in the process to efficiently scale to high multiplicities, and it guarantees the Lorentz invariance of the amplitude. Extending the studies performed in Ref. [35], we demonstrate its utility for the partonic processes

$$q\bar{q} \to Z + n\,g, \qquad n = 1\dots 5\,. \tag{20}$$

We train L-GATr networks with a standard MSE loss to predict the corresponding squared amplitudes $A$ from the initial and final state 4-momenta.

We generate $4 \times 10^5$ training data points for each multiplicity up to 4 gluons with Mad-

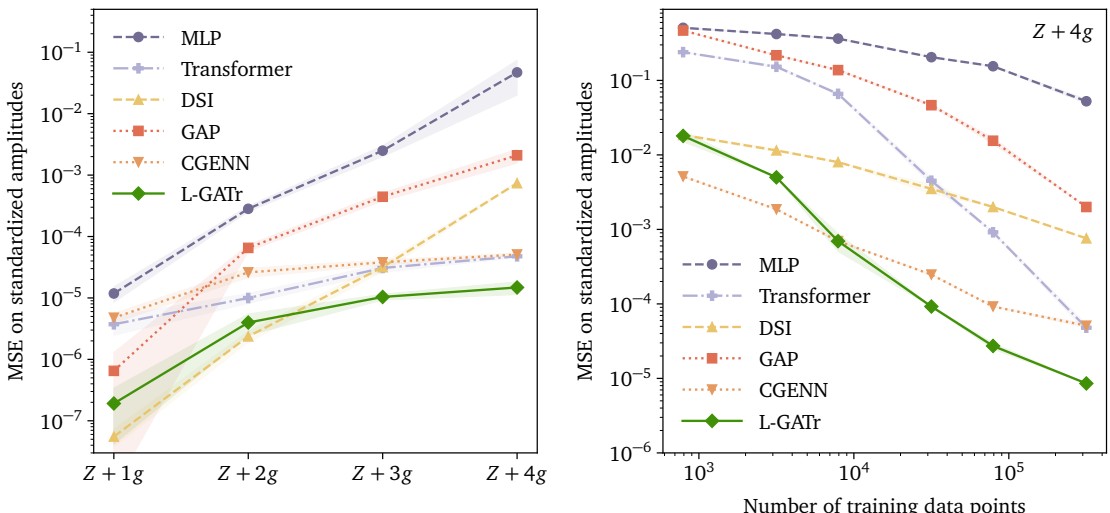

Figure 2: Left: prediction error from L-GATr and all baselines for $Z + ng$ amplitudes with increasing particle multiplicity. All networks are trained on $4\times10^5$ points. Right: prediction error as a function of the training dataset size. Error bands are based on the mean and standard deviation of five random seeds. These figures are also included in Ref. [35].

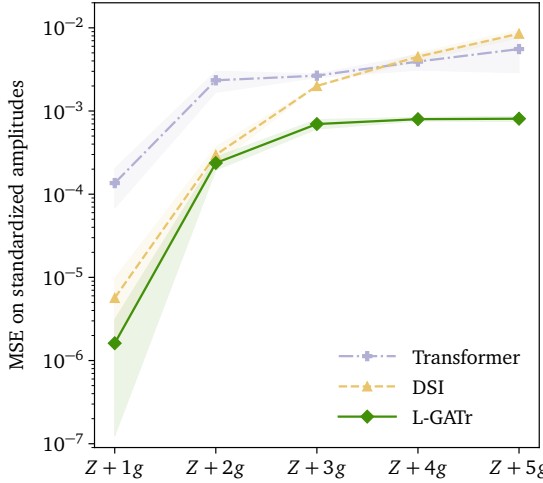

Figure 3: Prediction error from L-GATr and selected baselines for $Z + 5g$ amplitudes. Here, all networks are reduced in size and trained on $4 \times 10^4$ points. Error bands are based on the mean and standard deviation of five random seeds.

Graph [49]. First, we use a standard run to generate unweighted phase space points; second, we apply the standalone module to compute the squared amplitude values. To avoid divergences, we require globally

$$p_T > 20 \, \text{GeV} \qquad \text{and} \qquad \Delta R > 0.4 \tag{21}$$

for all final-state objects. We train on standardized logarithmic amplitudes

$$\mathcal{A} = \frac{\log A - \overline{\log A}}{\sigma_{\log A}} \, . \tag{22}$$

In addition to the L-GATr surrogate we also train a comprehensive set of benchmarks:

- a standard MLP;
- a standard transformer [38], as L-GATr without equivariance;
- the Geometric Algebra Perceptron (GAP), as L-GATr without transformer structure;
- a deep sets network (DSI) [50] combining partial Lorentz and permutation equivariance;
- the CGENN [14] as an equivariant graph network operating on multivectors like L-GATr.

Details about the implementation and training can be found in the Appendix A.

**Performance**

Using the MSE loss as a quality metric, we compare the performance of L-GATr to the baselines in the left panel of Fig. 2. Overall, transformer and graph networks scale better with the number of external particles. We find that L-GATr is roughly on par with the leading DSI network for a small number of gluons, but its improved scaling gives it the lead for higher-multiplicity final states. We find similar performance for L-GATr when we train a single network on all processes jointly.

Next, we show in the right panel of Fig. 2 how the performance scales with the size of the training dataset. L-GATr stands as a top performer on all training regimes. In particular for small training sets, L-GATr and CGENN are very efficient thanks to their equivariant operations.

To expand on the scaling with the number of gluons, we generate $4 \times 10^4$ points for $Z + 5$ gluon productions through the pipeline described above. Given this limited number of phase space points, we reduce the size of L-GATr to $4 \times 10^4$ parameters and the baselines to $10^4$ parameters, to prevent overfitting. In Fig. 3 we reproduce the L-GATr scaling behavior despite the more complex process and a less generous training.

## 4 L-GATr for Jet Tagging

Jet tagging is, arguably, the LHC task which is currently impacted most by modern ML. Two approaches stand out as top performers: transformer-based architectures and equivariant networks. In this section, we show how L-GATr sets a new record for jet tagging by combining the merits of both ideas. All results related to pre-training and multiclass tagging are new to this paper.

**Top tagging**

We first study the performance of L-GATr on the top tagging challenge [2], a representative and extensively studied jet tagging task at the LHC. The results in this section were first presented in Ref. [35].

The top tagging dataset, originally produced for Ref. [51], consists of 2 M top quark and QCD jets with

$$p_{T,j} = 550 \ldots 650 \, \text{GeV} \,, \tag{23}$$

generated with Pythia 8 [52] and interfaced with Delphes for detector simulation [53] using the default ATLAS card at that time. We train and evaluate the L-GATr tagger on this dataset following the standard train/validation/test splitting of 1.2/0.4/0.4 M. Details about the network implementation and the training method are provided in Appendix A.

We compare our L-GATr tagger with the following baselines:

| Network | Accuracy | AUC | $1/\epsilon_B$ ($\epsilon_S = 0.5$) | $1/\epsilon_B$ ($\epsilon_S = 0.3$) |
|---|---|---|---|---|
| TopoDNN [54] | 0.916 | 0.972 | – | 295 $\pm$ 5 |
| LoLa [9] | 0.929 | 0.980 | – | 722 $\pm$ 17 |
| $N$-subjettiness [55] | 0.929 | 0.981 | – | 867 $\pm$ 15 |
| PFN [56] | 0.932 | 0.9819 | 247 $\pm$ 3 | 888 $\pm$ 17 |
| TreeNiN [57] | 0.933 | 0.982 | – | 1025 $\pm$ 11 |
| ParticleNet [58] | 0.940 | 0.9858 | 397 $\pm$ 7 | 1615 $\pm$ 93 |
| ParT [59] | 0.940 | 0.9858 | 413 $\pm$ 16 | 1602 $\pm$ 81 |
| MIParT [60] | 0.942 | 0.9868 | 505 $\pm$ 8 | 2010 $\pm$ 97 |
| LorentzNet* [10] | 0.942 | 0.9868 | 498 $\pm$ 18 | 2195 $\pm$ 173 |
| CGENN* [14] | 0.942 | 0.9869 | 500 | 2172 |
| PELICAN* [42] | 0.9426 $\pm$ 0.0002 | 0.9870 $\pm$ 0.0001 | – | 2250 $\pm$ 75 |
| L-GATr* [35] | 0.9423 $\pm$ 0.0002 | 0.9870 $\pm$ 0.0001 | 540 $\pm$ 20 | 2240 $\pm$ 70 |
| ParticleNet-f.t. [60] | 0.942 | 0.9866 | 487 $\pm$ 9 | 1771 $\pm$ 80 |
| ParT-f.t. [60] | 0.944 | 0.9877 | 691 $\pm$ 15 | 2766 $\pm$ 130 |
| MIParT-f.t. [60] | 0.944 | 0.9878 | 640 $\pm$ 10 | 2789 $\pm$ 133 |
| L-GATr-f.t.* (new) | 0.9446 $\pm$ 0.0002 | 0.98793 $\pm$ 0.00001 | 651 $\pm$ 11 | 2894 $\pm$ 84 |

Table 2: Top tagging accuracy, AUC, and background rejection $1/\epsilon_B$ for the standard dataset [2, 51]. Lorentz-equivariant methods are indicated with an asterisk, and fine-tuning methods are separated with a horizontal line. Our error bars are based on the mean and standard deviation of five random seeds.

| Beam | Time | Embedding | AUC | $1/\epsilon_B$ ($\epsilon_S = 0.3$) |
|---|---|---|---|---|
| – | – | Token | 0.9844 | 1422 |
| $x_3^V = \pm 1$ | – | Token | 0.9850 | 1905 |
| – | $x_0^V = 1$ | Token | 0.9865 | 1923 |
| $x_{12}^B = x_{13}^B = x_{23}^B = 1$ | $x_0^V = 1$ | Token | 0.9863 | 2009 |
| $x_{12}^B = 1$ | $x_0^V = 1$ | Channel | 0.9865 | 2060 |
| $x_0^V = 1, x_3^V = \pm 1$ | $x_0^V = 1$ | Token | 0.9869 | 2114 |
| $x_3^V = \pm 1$ | $x_0^V = 1$ | Token | 0.9869 | 2152 |
| $x_{12}^B = 1$ | $x_0^V = 1$ | Token | 0.9870 | 2240 |

Table 3: Symmetry breaking. We compare the L-GATr performance on the top tagging dataset from Ref. [61] using different Lorentz symmetry breaking schemes. The last line is our default used in all other tagging experiments.

- LorentzNet [10], an equivariant graph network based on functions of the momentum invariants as coefficients for 4-momenta inputs;
- PELICAN [42], an alternative equivariant graph network based on momentum invariants and permutation equivariant aggregation functions;
- CGENN [14], an equivariant graph network operating on multivectors;
- ParT [59], a transformer that includes pairwise interaction features as an attention bias; and
- MIParT [60], an extension of ParT with specialized blocks that focus only on interaction features.

In Tab. 2, we see how L-GATr is at least on par with the leading equivariant baselines, as shown already in Ref. [35].

A key ingredient for the optimization of L-GATr is the symmetry breaking prescription. For all our tests, we include two reference vectors as extra tokens: the beam direction as the $x-y$ plane bivector, $x_{12}^B = 1$, and the time reference $x^V = (1, 0, 0, 0)$, which gives the network a handle to break the symmetry down to $SO(3)$. We provide a comparison between multiple reference vector options in Tab. 3, where we test their impact on a top tagging network. From it, it is clear that both the beam direction and the time reference significantly contribute to boosting the tagging performance.

**Multi-class tagging on JetClass**

We further study L-GATr for multiclass tagging with the JetClass dataset [59]. JetClass covers a wide variety of jet signatures. Its signal events consist of jets arising from multiple decay modes of top quarks, $W$, $Z$ and Higgs bosons; its background events are made up of light quark and gluon jets. All types of events are generated with MadGraph [62] and Pythia [52], and detector effects are simulated with Delphes [63] using the default CMS card. A kinematic cut

$$p_{T,j} = 500...1000 \text{ GeV} \qquad \text{and} \qquad |\eta_j| < 2.0 \qquad (24)$$

is applied to all jets in the dataset. In total, JetClass contains 100 M jets equally distributed across 10 classes.

JetClass provides input features of four main categories: the 4-momenta of the jet particles, kinematic variables like $\Delta R$ and $\log p_T$ that can be derived from the 4-momenta, particle identification variables, and trajectory displacement variables. When passing them through L-GATr, all features besides the 4-momenta are embedded to the network as scalar channels. We use the same architecture as we did for top tagging in all our tests (see Appendix A).

| | All classes | | $H \to b\bar{b}$ | $H \to c\bar{c}$ | $H \to gg$ | $H \to 4q$ | $H \to l\nu q\bar{q}'$ | $t \to bq\bar{q}'$ | $t \to bl\nu$ | $W \to q\bar{q}'$ | $Z \to q\bar{q}$ |
| | Accuracy | AUC | Rej$_{50\%}$ | Rej$_{50\%}$ | Rej$_{50\%}$ | Rej$_{50\%}$ | Rej$_{99\%}$ | Rej$_{50\%}$ | Rej$_{99.5\%}$ | Rej$_{50\%}$ | Rej$_{50\%}$ |
|---|---|---|---|---|---|---|---|---|---|---|---|
| ParticleNet [58] | 0.844 | 0.9849 | 7634 | 2475 | 104 | 954 | 3339 | 10526 | 11173 | 347 | 283 |
| ParT [59] | 0.861 | 0.9877 | 10638 | 4149 | 123 | 1864 | 5479 | 32787 | 15873 | 543 | 402 |
| MIParT [60] | 0.861 | 0.9878 | 10753 | 4202 | 123 | 1927 | 5450 | 31250 | 16807 | 542 | 402 |
| L-GATr | 0.866 | 0.9885 | 12987 | 4819 | 128 | 2311 | 6116 | 47619 | 20408 | 588 | 432 |

Table 4: Tagging accuracy, AUC, and background rejection $1/\epsilon_B$ for the JetClass dataset [59]. The AUC is computed as the average of all possible pairwise combinations of classes, and the acceptances are computed by comparing each signal against the background class.

We present the results in Tab. 4, including reference metrics from Refs. [59, 60]. The L-GATr tagger achieves a significant improvement over the previous state-of-the-art, ParT and MIParT, in essentially all signal types. In a separate study, we have checked that the quality of L-GATr predictions steadily increases as we add more features to the training data. We also train L-GATr with subsets of the dataset to test its data efficiency. As we can see in the left panel in Fig. 4 and Table 5, L-GATr achieves a performance similar to the non-equivariant ParT and MIParT taggers even if trained with only 10% of all available jets.

**Top tagging with JetClass pre-training**

The large capacity of transformer architectures motivates using pre-training to further improve the tagging performance [59]. To this end, we pre-train L-GATr on the JetClass dataset and fine-tune it for top tagging. The pre-training uses the same setup as the JetClass training, but the inputs are limited to the 4-momenta and the derived kinematic variables, as those are the only available features in the top tagging dataset.

We follow Ref. [59] for the pre-training and fine-tuning procedures. Once the network is pre-trained on the large dataset, we fine-tune it by switching the last layer of the network to map to a single output channel and re-initialize its weights. During fine-tuning, the pre-trained model weights have to be updated with a smaller learning rate than the new ones, otherwise the network might dismiss all information from the pre-training. Further details about the

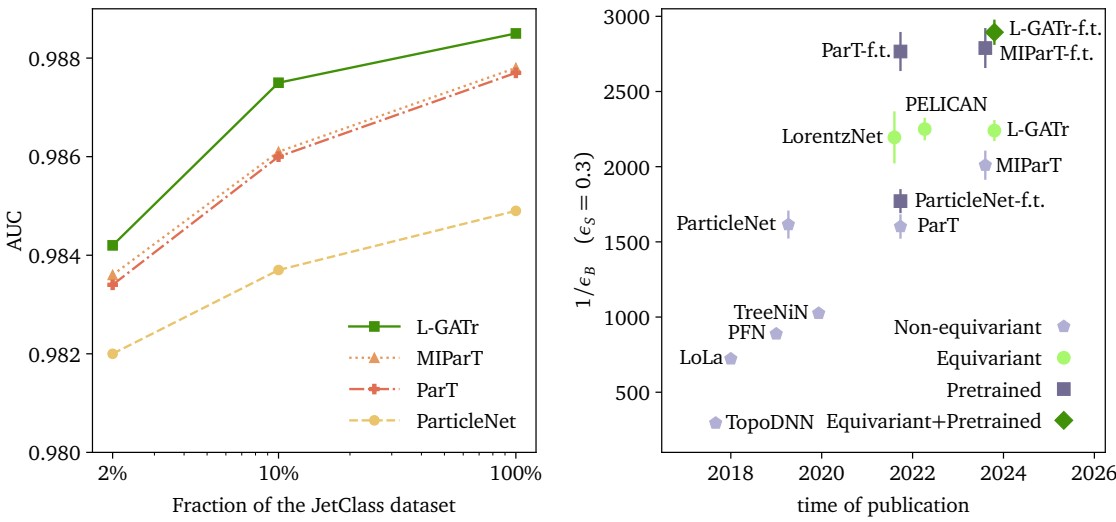

Figure 4: AUC metric on JetClass as a function of the training dataset fraction (left) and the history of top taggers (right).

| | All classes | | $H \to b\bar{b}$ | $H \to c\bar{c}$ | $H \to gg$ | $H \to 4q$ | $H \to l\nu q\bar{q}'$ | $t \to bq\bar{q}'$ | $t \to bl\nu$ | $W \to q\bar{q}'$ | $Z \to q\bar{q}$ |
| | Accuracy | AUC | Rej$_{50\%}$ | Rej$_{50\%}$ | Rej$_{50\%}$ | Rej$_{50\%}$ | Rej$_{99\%}$ | Rej$_{50\%}$ | Rej$_{99.5\%}$ | Rej$_{50\%}$ | Rej$_{50\%}$ |
|---|---|---|---|---|---|---|---|---|---|---|---|
| ParticleNet (2 M) | 0.828 | 0.9820 | 5540 | 1681 | 90 | 662 | 1654 | 4049 | 4673 | 260 | 215 |
| ParticleNet (10 M) | 0.837 | 0.9837 | 5848 | 2070 | 96 | 770 | 2350 | 5495 | 6803 | 307 | 253 |
| ParticleNet (100 M) | 0.844 | 0.9849 | 7634 | 2475 | 104 | 954 | 3339 | 10526 | 11173 | 347 | 283 |
| ParT (2 M) | 0.836 | 0.9834 | 5587 | 1982 | 93 | 761 | 1609 | 6061 | 4474 | 307 | 236 |
| ParT (10 M) | 0.850 | 0.9860 | 8734 | 3040 | 110 | 1274 | 3257 | 12579 | 8969 | 431 | 324 |
| ParT (100 M) | 0.861 | 0.9877 | 10638 | 4149 | 123 | 1864 | 5479 | 32787 | 15873 | 543 | 402 |
| MIParT (2 M) | 0.837 | 0.9836 | 5495 | 1940 | 95 | 819 | 1778 | 6192 | 4515 | 311 | 242 |
| MIParT (10 M) | 0.850 | 0.9861 | 8000 | 3003 | 112 | 1281 | 3650 | 16529 | 9852 | 440 | 336 |
| MIParT (100 M) | 0.861 | 0.9878 | 10753 | 4202 | 123 | 1927 | 5450 | 31250 | 16807 | 542 | 402 |
| L-GATr (2 M) | 0.839 | 0.9842 | 6623 | 2294 | 99 | 981 | 1980 | 8097 | 4902 | 346 | 276 |
| L-GATr (10 M) | 0.859 | 0.9875 | 9804 | 3883 | 120 | 1791 | 4255 | 24691 | 13333 | 506 | 373 |
| L-GATr (100 M) | 0.866 | 0.9885 | 12987 | 4819 | 128 | 2311 | 6116 | 47619 | 20408 | 588 | 432 |

Table 5: Tagging accuracy, AUC, and background rejection $1/\epsilon_B$ on different sizes of the JetClass dataset [59]. Metrics from other models are taken from their published results [59, 60].

fine-tuning setup are discussed in Appendix A.

We show the results from fine-tuned L-GATr in Tab. 2, where we compare different fine-tuned baselines. L-GATr matches the performance of the best fine-tuned networks in the literature across all metrics.[‡] To further illustrate the impact of combining equivariance and pre-training, we summarize the historical progress in top tagging in the right panel of Fig. 4.

# 5 L-GATr for Event Generation

Generating LHC events is a key benchmark for neural network architectures, required for end-to-end generation, neural importance sampling, generative unfolding [64–69], and optimal inference [70, 71]. For all these tasks we should reach per-mille-level, or at the very least percent-level accuracy on the underlying phase space density. We use the L-GATr architecture to take advantage of the approximate symmetries and to improve the scaling for increasing numbers of final state particles. Our reference process is

$$pp \to t_h \bar{t}_h + n\,j, \qquad n = 0 \ldots 4\,, \qquad (25)$$

with both top quarks decaying hadronically. It is simulated with MadGraph3.5.1, consisting of MadEvent [72] for the underlying hard process, Pythia8 [52] for the parton shower, Delphes3 [53] for the detector simulation, and the anti-$k_T$ jet reconstruction algorithm [73] with $R = 0.4$ as implemented in FastJet [74]. We use Pythia without multi-parton interactions and the default ATLAS detector card. We apply the phase space cuts

$$p_{T,j} > 22 \text{ GeV} \qquad \Delta R_{jj} > 0.5 \qquad |\eta_j| < 5\,, \qquad (26)$$

and require two $b$-tagged jets. The events are reconstructed with a $\chi^2$-based algorithm [75], and identical particles are ordered by $p_T$. The sizes of the $t\bar{t} + n\,j$ datasets reflect the frequency of the respective processes, resulting in 9.8M, 7.2M, 3.7M, 1.5M and 480k events for $n = 0 \ldots 4$. We train separate networks for each multiplicity, to allow for a direct comparison between different architectures, but we emphasize that transformers can also be trained jointly on all multiplicities [76]. This is particularly useful in the case of limited training data, because transformers can transfer information across multiplicities. The results presented here were briefly discussed in Ref. [35], but without proper benchmarking.

---

[‡]In this table, we see that the prediction made in the title of Ref. [8] turned out accurate.

**Conditional flow matching (CFM)**

Continuous normalizing flows [77] learn a continuous transition $x(t)$ between a simple latent distribution $x_1 \sim p_{\text{latent}}(x_1)$ and a phase space distribution $x_0 \sim p_{\text{data}}(x_0)$. Mathematically, they build on two equivalent ways of describing a diffusion process, using either an ODE or a continuity equation [1, 76]

$$\frac{dx(t)}{dt} = v(x(t), t) \qquad \text{or} \qquad \frac{\partial p(x, t)}{\partial t} = -\nabla_x [v(x(t), t)p(x(t), t)] , \qquad (27)$$

with the same CFM-velocity field $v(x(t), t)$. The diffusion process $t = 0 \to 1$ interpolates between the phase space distribution $p_{\text{data}}(x_0)$ and the base distribution $p_{\text{latent}}(x_1)$,

$$p(x, t) \to \begin{cases} p_{\text{data}}(x_0) & t \to 0 \\ p_{\text{latent}}(x_1) & t \to 1 . \end{cases} \qquad (28)$$

To train the continuous normalizing flow with conditional flow matching [78, 79], we employ a simple linear interpolation

$$x(t) = (1-t)x_0 + tx_1 \to \begin{cases} x_0 & t \to 0 \\ x_1 & t \to 1 . \end{cases} \qquad (29)$$

and train the network with parameters $\theta$ on a standard MSE loss to encode the (CFM-)velocity

$$v_\theta \left( (1-t)x_0 + tx_1, t \right) \approx x_1 - x_0. \qquad (30)$$

The network has to learn to match the un-conditional velocity field on the left hand side to the conditional velocity field on the right hand side. We then generate phase space configurations using a fast ODE solver via

$$x_0 = x_1 - \int_0^1 dt \, v_\theta(x(t), t). \qquad (31)$$

**L-GATr velocity**

We compare the L-GATr performance with a set of leading benchmark architectures, all based on a CFM generator, with

- a standard MLP [76];
- a standard transformer [69]; and
- an E(3)-GATr [33].

They share the generative CFM setup, which is currently the leading technique in precision generation of partonic LHC events [76] and calorimeter showers, for the latter using a vision transformer inside the CFM [80]. Our task does not require translation-equivariant representations. Therefore, we do not insert or extract features related to translational equivariance when using the E(3)-GATr, but internally it can use these representations. Details about the implementation and training can be found in the Appendix A.

Strictly speaking, the underlying problem is only symmetric under rotations around the beam axis. To not over-constrain the network we include symmetry breaking multivectors as described in Sec. 2.3. For the E(3)-GATr, we include the plane orthogonal to the beam axis as a reference multivector. For L-GATr, we also include a time reference multivector, because the distribution is not invariant under boosts along the beam axis. This step is necessary for any

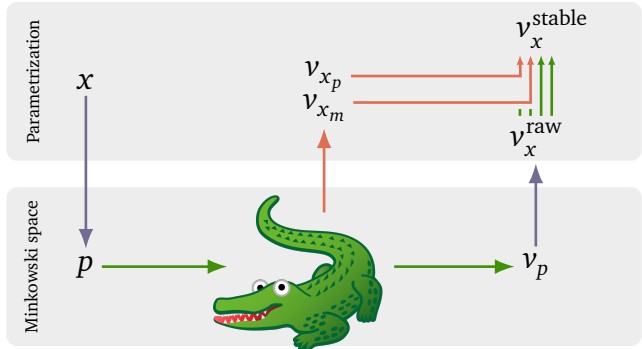

Figure 5: To construct the L-GATr velocity, we extract equivariantly predicted multivectors and symmetry-breaking scalars. We go back and forth between the parametrization $x$ and Minkowski space $p$ using the mapping $f$ from Eq. (32).

Lorentz-equivariant generative network, as it is not possible to construct a normalized density that is invariant under a non-compact group.

To construct an equivariant generator, we have to choose a base distribution $p_{\text{latent}}(x_1)$ that is invariant under the symmetry group. We use gaussian distributions in the coordinates $(p_x, p_y, p_z, \log m^2)$ with mean and standard deviation fitted to the phase space distribution $p_{\text{data}}(x_0)$. Furthermore, we apply rejection sampling to enforce the phase space constraints $p_T > 22\,\text{GeV}, \Delta R > 0.5$ already at the level of the base distribution.

The phase space parametrization for which we require straight trajectories is crucial for the performance of the generator. The standard MLP and transformer CFMs work directly on $x$ defined as

$$p = \begin{pmatrix} E \\ p_x \\ p_y \\ p_z \end{pmatrix} \quad \rightarrow \quad f^{-1}(p) = x = \begin{pmatrix} x_p \\ x_m \\ x_\eta \\ x_\phi \end{pmatrix} \equiv \begin{pmatrix} \log(p_T - p_T^{\min}) \\ \log m^2 \\ \eta \\ \phi \end{pmatrix}, \tag{32}$$

to encode $v(x(t), t)$. We standardize all four $x$-coordinates using their mean and standard deviation over the full dataset. The azimuthal angle $\phi$ is periodic, and we use this property by adding multiples of $2\pi$ to map generated angles into the allowed region $\phi \in [-\pi, \pi]$. We then choose the smallest distance between pairs $(x_0, x_1)$ to construct the target velocity field, allowing paths to cross the boundary at $\phi = \pm\pi$.

L-GATr starts with $p$ and applies the transformation visualized in Fig. 5: first, we use the mapping $f$ to transform $x$ into the corresponding 4-momentum $p = f(x)$. Second, we apply the L-GATr network to obtain the velocity $v_p = \text{L-GATr}(p) = (v_E, v_{p_x}, v_{p_y}, v_{p_z})$ in Minkowski

| Data | Architecture | Base distribution | Periodic | Neg. log-likelihood | AUC |
|------|-------------|-------------------|----------|---------------------|-----|
| $p$ | L-GATr | rejection sampling | ✓ | -30.80 $\pm$ 0.17 | 0.945 $\pm$ 0.004 |
| $x$ | MLP | rejection sampling | ✓ | -32.13 $\pm$ 0.05 | 0.780 $\pm$ 0.003 |
| $x$ | L-GATr | rejection sampling | ✗ | -32.57 $\pm$ 0.05 | 0.530 $\pm$ 0.017 |
| $x$ | L-GATr | no rejection sampling | ✓ | -32.58 $\pm$ 0.04 | 0.523 $\pm$ 0.014 |
| $x$ | L-GATr | rejection sampling | ✓ | -32.65 $\pm$ 0.04 | 0.515 $\pm$ 0.009 |

Table 6: Impact of the choice of trajectory on different L-GATr networks for the CFM velocity, compared to a MLP velocity network. All networks are trained on the $t\bar{t}+0j$ dataset.

space. Finally, we transform this velocity $v_p$ back into the parametrization $x$ using the jacobian of the backwards transformation, yielding the transformed velocity $v_x = (v_{x_p}, v_{x_m}, v_\eta, v_\phi)$

$$v_x(x(t), t) = \frac{\partial f^{-1}(p)}{\partial p} v_p(p(t), t). \tag{33}$$

In practice, we encounter large jacobians from the logarithm transformations in the above relation for $v_{x_m}, v_{x_p}$ due to small values of the jet mass $m$ and the relative transverse momentum $p_T - p_{T,\min}$, leading to unstable training. To avoid this obstacle, we add a fourth step to the procedure, where we overwrite the two problematic velocity components with scalar outputs of the L-GATr network. This adds an additional redundant source of symmetry breaking to the reference multivectors discussed above.

For the E(3)-GATr, we encode $(p_x, p_y, p_z)$ as a vector and $x_m$ as a scalar. We then apply a similar transformation as for L-GATr, but without changing the $x_m$ component.

For the full L-GATr architecture we first study the effect of the choice of data representation, of base distribution, and of trajectories in Tab. 6.

**Performance**

We compare a set of 1-dimensional distributions from the different generators in Fig. 6. Observables like $p_{T,b}$, that are part of the phase space parametrization Eq. (32), are easily learned by all networks because of our choice of target trajectories in Eq. (29). All other observables appear s correlations and are harder to learn. L-GATr outperforms the baselines across all distributions. Especially angular correlations benefit from the equivariance encoded in the L-GATr architecture, enabling percent-level precision in these variables for the first time. The main weakness of all architectures are the intermediate top mass poles, requiring the correlation of three external 4-vectors.

To analyze scaling properties, we use the negative log-likelihood evaluated on the events and the AUC of a neural classifier. In Fig. 7 we find a clear performance increase with increasing symmetry awareness, from the unstructured MLP over the permutation-equivariant transformer to the rotation-equivariant GATr and the Lorentz-equivariant L-GATr. In particular, the superior L-GATr performance mainly originates from boost-equivariance, as the rotation-equivariant E(3)-GATr performs only marginally better than the plain transformer. This might come as a surprise, as we allow L-GATr to break this boost equivariance using reference multivectors. This implies that enforcing equivariance in the architecture and then allowing the network to break it with reference multivectors outperforms standard non-equivariant networks.

# 6 Outlook

Modern ML at the LHC has developed from mostly concept development to the first applications in experiment and theory. For these applications, performance is the main goal, leading us to the question how we can train the most precise neural networks on a large, but nevertheless limited training dataset. In LHC physics, we are in the lucky situation that we can use the known structure of the phase space. It rests on a complex system of symmetries, the leading one being the Lorentz symmetry.

To help our network training, we can encode the Lorentz symmetry or Minkowski metric into the network architecture to avoid learning it. An appropriate internal or latent representation of the Lorentz group then enhances the performance of, essentially, every ML-application working on relativistic phase space objects. Crucially, in cases where symmetries are not exact, we can allow an equivariant network to break them using symmetry-breaking reference

frames, leading to significantly better performance than removing the corresponding equivariance from the network altogether.

L-GATr is a versatile equivariant transformer that constructs such a Lorentz representation

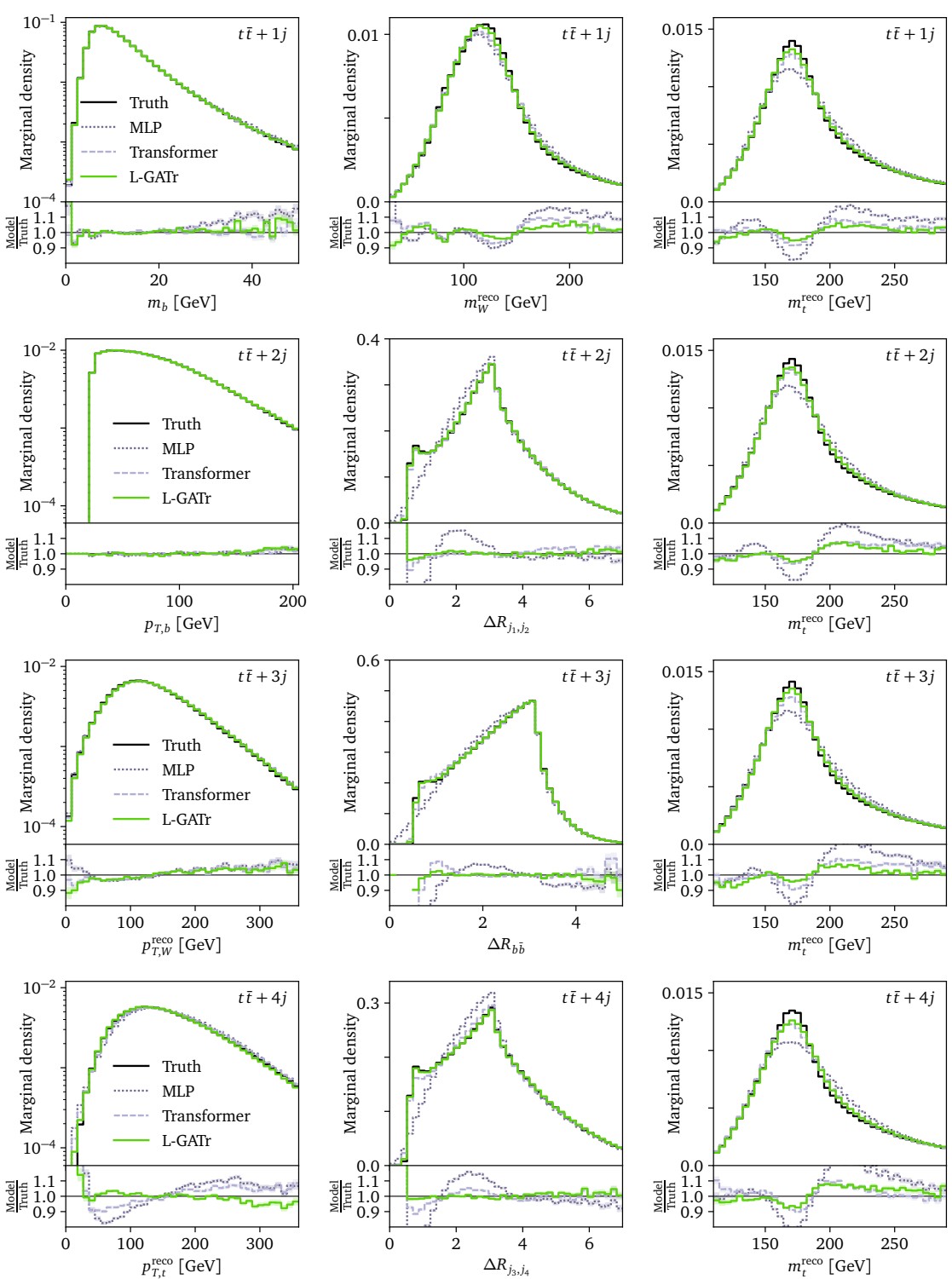

Figure 6: Marginal distributions for $t\bar{t} + 1, 2, 3, 4$ jets (top to bottom). We do not show E(3)-GATr results, as they are very similar to the standard transformer. The three panels in the bottom row are also included in Ref. [35].

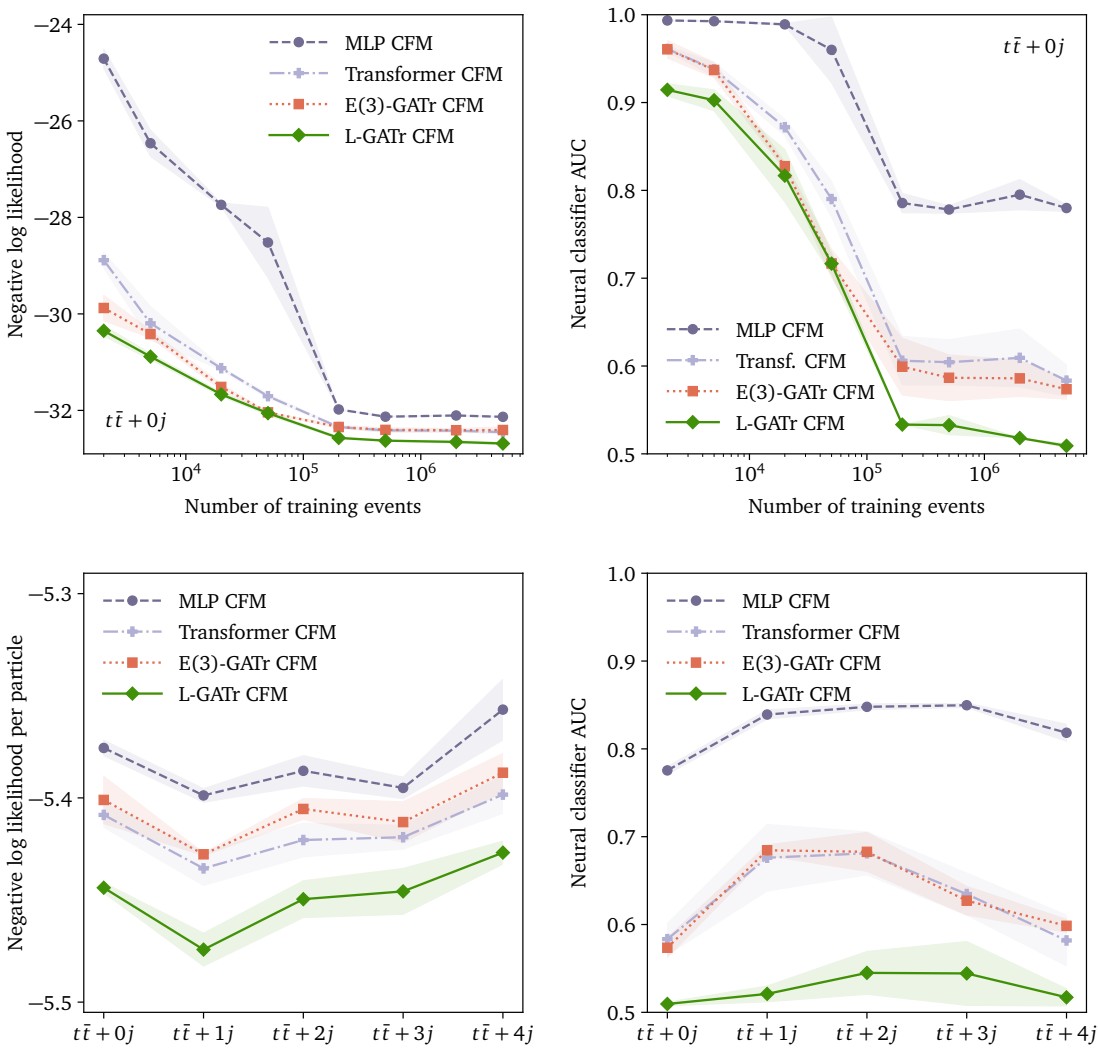

Figure 7: Performance of generative networks in terms of a negative log-likelihood over the events (left) and a trained classifier AUC (right). In the top row we show the scaling with the size of the training dataset for the mixed $t\bar{t} + n$ jet dataset, in the bottom row the scaling with the number of particles in the final state. The MLP, Transformer and L-GATr metrics were already discussed in Ref. [35].

for regression, classification, and generation networks. For amplitude regression, a key step in speeding up loop amplitudes in event generators, L-GATr shows the best performance for more than three particles in the final state, thanks to its superior data efficiency, leading to an improved scaling with the phase space dimensionality. For subjet tagging, L-GATr combines the benefit of equivariance with pre-training on large datasets and is at least on par with the best available subjet tagger. Finally, the combination of L-GATr with CFM generator faithfully reproduces the phase space distribution of top pair production with up to four jets better than all other CFM setups. We look forward to further applications of L-GATr at the LHC, as well as generalizations to incorporate additional domain knowledge from collider physics.

**Code availability**

L-GATr is available at https://github.com/heidelberg-hepml/lorentz-gatr as part of the public Heidelberg hep-ml code and tutorial library.

**Acknowledgements**

This paper benefited from great discussions with Taco Cohen at the *2023 Hammers & Nails* conference. We would like to thank Anja Butter, David Ruhe, and Nathan Hütsch for many useful discussions. This work was supported by the by the DFG under grant 396021762 – TRR 257: *Particle Physics Phenomenology after the Higgs Discovery*, and through Germany's Excellence Strategy EXC 2181/1 – 390900948 (the *Heidelberg STRUCTURES Excellence Cluster*). The authors acknowledge support by the state of Baden-Württemberg through bwHPC and the German Research Foundation (DFG) through grant INST 35/1597-1 FUGG. J.S. is funded by the Carl-Zeiss-Stiftung through the project *Model-Based AI: Physical Models and Deep Learning for Imaging and Cancer Treatment*. V.B. is supported by the BMBF Junior Group *Generative Precision Networks for Particle Physics* (DLR 01IS22079). V.B. acknowledges financial support from the Grant No. ASFAE/2022/009 (Generalitat Valenciana and MCIN, NextGenerationEU PRTR-C17.I01). J.T. is supported by the National Science Foundation under Cooperative Agreement PHY-2019786 (The NSF AI Institute for Artificial Intelligence and Fundamental Interactions, http://iaifi.org/), by the U.S. Department of Energy Office of High Energy Physics under grant number DE-SC0012567, and by the Simons Foundation through Investigator grant 929241.

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

## A Network and Training Details

**Amplitude regression**

Concerning the DSI baseline, it is an architecture based on the Deep Sets framework [50] that incorporates momentum invariants as part of the input. It works in three stages. First, it applies a different learnable preprocessing block to each particle type in the events, generating a set of latent space representations for each of the particle inputs. Those latent space points are then combined way by summing over all identical particle types, effectively imposing permutation invariance. Finally, the resulting aggregation together with a collection of all momentum invariants of the process is fed to another block that performs the actual regression task. This setup achieves a combination of Lorentz and permutation invariants in an imperfect way.

We list the hyperparameters of all studied baselines in Table 7. As for the preprocessing, in the case of GAP and L-GATr we standardize the 4-momentum inputs using a common normalization for each component to preserve Lorentz equivariance. For the rest of the baselines we perform ordinary standarization.

**Jet Tagging**

We provide the L-GATr hyperparamters for top tagging without pre-training in Table 8. All inputs are preprocessed with a 20 GeV scale factor. L-GATr is trained by minimizing a binary cross entropy loss on the class labels.

Pre-training and full training on JetClass is performed by training L-GATr on the full 100M events over $10^6$ iterations. The L-GATr architecture and training hyperparameters is the same

that we used for the ordinary top tagging. The only differences are that we now work with 10 output channels and train on a cross entropy loss to accommodate multiclass training, and we use a batch size of 512 to maximize dataset exposure. As for the inputs, the 4-momenta are again scaled by the 20 GeV scale, and the kinematic functions are standardized following the prescription presented in Ref. [59].

Fine-tuning is implemented by resetting the output layer of the pre-trained network and restricting it to one output channel. With this build, the pre-trained weights are trained with a learning rate of $3 \times 10^{-5}$ and the new weights are trained with a learning rate of $3 \times 10^{-3}$. We also apply a weight decay of 0.01 and a batch size of 128. The training is performed across $10^5$ iterations.

**Event Generation**

We summarize the architecture and training hyperparameters of all four generator baselines in Table 9. We split each dataset into 98% for training and 1% each for validation and testing.

In the classifier test, we train an MLP classifier using binary cross-entropy to distinguish generated events from true events. The classifier inputs include the complete events in the $x$ representation defined in Eq. (32), augmented by all pairwise $\Delta R$ features, as well as the $x$ representations of the reconstructed particles $t, \bar{t}, W^+, W^-$. The classifier network consists of 3 layers with 256 channels each. Training is conducted over 500 epochs with a batch size of 1024, a dropout rate of 0.1, and the Adam optimizer with default parameters. We start with an initial learning rate of 0.0003, reducing it by a factor of 10 if validation loss shows no improvement for 5 consecutive epochs. The dataset comprises the full truth data and 1M generated events, with an 80%/10%/10% split for training, testing, and validation, respectively.

| Hyperparameter | MLP | DSI | Transformer | GAP | CGENN | L-GATr |
|---|---|---|---|---|---|---|
| Architecture | 128 channels 5 layers | 128 channels 4 layers | 128 channels 8 heads 8 blocks | 96 scalar ch. 96 multivector ch. 8 blocks | 72 scalar ch. 8 multivector ch. 4 blocks | 32 scalar ch. 32 multivector ch. 8 heads 8 blocks |
| Activation | GELU | GELU | GELU | Gated GELU | Gated SiLU [14] | Gated GELU |
| Parameters | $7 \times 10^4$ | $2.6 \times 10^5$ | $1.3 \times 10^6$ | $2.5 \times 10^6$ | $3.2 \times 10^5$ | $1.8 \times 10^6$ |
| Optimizer | Adam [81] | Adam [81] | Adam [81] | Adam [81] | Adam [81] | Adam [81] |
| Learning rate | $10^{-4}$ | $10^{-4}$ | $10^{-4}$ | $10^{-4}$ | $10^{-4}$ | $10^{-4}$ |
| Batch size | 256 | 256 | 256 | 256 | 256 | 256 |
| Scheduler | - | - | - | - | - | - |
| Patience | 100 | 100 | 100 | 100 | 100 | 100 |
| Iterations | $2.5 \times 10^6$ | $2.5 \times 10^6$ | $10^6$ | $2.5 \times 10^5$ | $2.5 \times 10^5$ | $2.5 \times 10^5$ |

Table 7: Hyperparameter summary for all baselines studied for the amplitude task. In the case of DSI, the number of layers and channels refers to both network blocks, and the latent space of each particle has a dimensionality of 64. In the case of CGENN, the hidden node features are identified with the scalar channels, whereas the hidden edge features are identified with the multivector channels.

| Hyperparameter | Value |
|---|---|
| Scalar channels | 32 |
| Multivector channels | 16 |
| Attention heads | 8 |
| Blocks | 12 |
| Parameters | $1.1 \times 10^6$ |
| Optimizer | Lion [82] |
| Learning rate | $3 \times 10^{-4}$ |
| Batch size | 128 |
| Scheduler | CosineAnnealingLR [83] |
| Weight decay | 0.2 |
| Iterations | $2 \times 10^5$ |

Table 8: Hyperparameter summary for the L-GATr network used for top tagging.

| Hyperparameter | MLP | Transformer | L-GATr and E(3)-GATr |
|---|---|---|---|
| Architecture | 336 <br> 6 layers | 108 channels <br> 6 layers <br> 8 heads | 32 scalar ch. <br> 16 multivector ch. <br> 8 heads <br> 6 blocks |
| Activation | GELU | GELU | Gated GELU |
| Parameters | $5.9 \times 10^5$ | $5.7 \times 10^5$ | $5.4 \times 10^5$ |
| Optimizer | Adam [81] | Adam [81] | Adam [81] |
| Learning rate | $10^{-3}$ | $10^{-3}$ | $10^{-3}$ |
| Batch size | 2048 | 2048 | 2048 |
| Iterations | $2 \times 10^5$ | $2 \times 10^5$ | $2 \times 10^5$ |

Table 9: Hyperparameter summary for all baselines studied for the generation task. For all networks, we evaluate the validation loss every $10^3$ iterations and decrease the learning rate by a factor of 10 after no improvements for 20 validation steps.