# Peer review of "A Lorentz-Equivariant Transformer for All of the LHC"

_SciPost Physics_

## Round 2 · Referee Report · Anonymous (Referee 2) · 2025-5-7

Strengths

  1. The paper introduces a novel Neural Network architecture — Lorentz-Equivariant Geometric Algebra Transformer (L-GATr) — which leverages geometric algebra to ensure Lorentz equivariance, marking a significant conceptual advancement in the design of symmetry-aware models for particle physics.

  2. The proposed method is universal in scope, demonstrating its versatility by being successfully applied to three distinct and relevant tasks: amplitude regression, jet tagging, and event generation.

  3. Across all tasks, the model achieves competitive or superior performance compared to existing architectures, including state-of-the-art benchmarks, thereby validating its practical effectiveness.

Weaknesses

  1. A significant portion of the results presented in the manuscript have already appeared in a previous work by the authors, titled "Lorentz-Equivariant Geometric Algebra Transformers for High-Energy Physics" (arXiv:2405.14806 [physics.data-an]). This raises concerns regarding the degree of novelty in the current submission.

  2. The explanation of the top-tagging task lacks clarity and could benefit from a more structured and detailed presentation.

  3. The manuscript offers limited insight into the sources of performance improvement, with little analysis devoted to understanding why the proposed model outperforms others.

  4. The fairness of the comparisons made between different architectures is questionable, particularly due to variations in the number of trainable parameters.

Report

The authors present a novel Neural Network architecture: the Lorentz-Equivariant Geometric Algebra Transformer (L-GATr). Its principal innovation lies in the representation of data within a geometric algebra over space-time, combined with Lorentz equivariance. Furthermore, the architecture permits the controlled breaking of this symmetry, which is particularly beneficial for collider physics applications. The proposed algorithm is versatile and may be applied to a broad range of problems in particle physics. The authors evaluate its performance on three tasks: amplitude regression, jet tagging, and event generation.

In the amplitude regression task, the authors consider the process qq~ -> Z + ng (with n = 1,...,5) and compare the performance of L-GATr against several existing architectures. L-GATr achieves the lowest Mean Squared Error for n = 3, 4, 5, and results comparable to the Deep Sets Network for n = 1 and n = 2. For jet tagging, the architecture is first tested on a standard QCD vs. top-jet dataset, where its performance matches that of the state-of-the-art PELICAN network. Subsequently, L-GATr is trained on the JetClass dataset, which comprises 100 million events equally distributed across ten jet classes. In this setting, L-GATr outperforms all other tested models. The network trained on JetClass is then fine-tuned for top tagging, where it surpasses PELICAN and establishes itself as the new state-of-the-art.

In the final task - event generation — the authors focus on the process pp -> t t~ + n j (with n = 0,...,4), assuming hadronic top decays. The results demonstrate excellent agreement between the generated data and events simulated with MadGraph and Pythia.

Overall, the presented results are compelling and address important challenges in collider physics, representing a significant step toward the development of a foundational model. While the manuscript is generally well written, I would like to highlight several concerns, which I outline below. Due to these issues, I ask for a major revision.

Requested changes

1) The most pressing issue relates to the overlap between the current submission and a previously released preprint by the same authors, entitled "Lorentz-Equivariant Geometric Algebra Transformers for High-Energy Physics" (arXiv:2405.14806 [physics.data-an]). Although the authors are transparent in referencing this earlier work and acknowledging the reuse of results, concerns regarding the novelty of the present submission naturally arise. Specifically, the following elements appear to have been reproduced from the preprint: the left panel of Fig. 1, Fig. 2, the top-tagging result in Table 2, the bottom row of Fig. 6, and Fig. 7. The new results include the right panel of Fig. 2, Fig. 3, the fine-tuned top-tagging results in Table 2, JetClass tagging results in Tables 4 and 5, Fig. 4, and several new histograms in Fig. 6. However, among these, only the JetClass tagging and fine-tuned top-tagging results appear to constitute substantial novel contributions; the remaining additions could reasonably be regarded as supplementary material.

In light of this, I respectfully request that the authors clarify the status of the earlier document, whether it is intended as an online supplement, a preprint, or a separate publication. I would also encourage the authors to explicitly articulate the novelty of their submission to SciPost in order to facilitate a more accurate evaluation of its contribution to the field.

Below, I list other comments. Whenever clarifying something, please address it also in the article text. If something is not clear to me, it might also be unclear for other readers.

2) Regarding Eq. 10 and Eq. 11: I believe there is a factor of 1/2 missing in Eq. 10 if it is to be consistent with Eq. 11. Please recalculate.

3) Regarding Eq. 18: The notation in the RHS suggests that the whole multivector is scaled by GELU(<x>_0). Is that what is really done? Please correct or clarify.

Regarding Sec. 2.4:

4) "We execute the network forward passes with synthetic data on an H100 GPU." -> Why do you use synthetic data instead of real one? What kind of synthetic data is it? What do you mean by executing the network forward pass? Generation of tokens by an already trained model? Please clarify.

5) "To ensure fairness, all networks consist of a single network block (...)" -> What about the number of parameters? Is it similar or different? What about optimiser, number of epochs, etc? It can affect the fairness of the comparison. Please clarify and justify.

Regarding Sec 3:

6) When reading this section for the first time, I was confused. You write about studying n=1..5 in text, then in Fig. 2, the n=5 is missing. Please clarify the description in that section.

7) Why did you decide to extend to n=5 only for the smaller network, with fewer data points, and compare it with only 2 other models? I would expect an extension of Fig.2 left, but it is not done. Please update the result or properly justify why it has not been done.

8) In Fig.2 and other figures, the results have a 1-sigma uncertainty band calculated using 5 random seeds. What do these seed values affect? Is it only the weight initialisation, also train-val-test splitting, or data generation (for section 5)? Please clarify.

9) Figs. 2 and 3 show different results. For example, in Fig. 2, L-GATr is worse than DSI for n=1, but in Fig. 3, it is better. For n=4 in Fig. 2, the MSE for the Transformer and DSI differ by an order of magnitude. However, in Fig. 3, they give the same result. Do you have an idea why it is like that? Do you think that the bands might be underestimating the true uncertainty? The difference between the two plots is not addressed in the text, and should at least be mentioned.

10) Regarding Tab. 2: Please clarify the origin of the values in the table. For example, the metric values for the LoLa network come from "The Machine Learning landscape of top taggers", Kasieczka et al., SciPost Phys. 7, 014 (2019), which is reference [2] in the manuscript. However, a reader might think that these values are from the original LoLa paper, i.e. reference [9]. By the way, the missing values (but not the uncertainties) of 1/eps_B for eps_S=0.5 can be read from Fig. 5 in [2].

11) Regarding Sec. 4: "A key ingredient for the optimisation of L-GATr is the symmetry-breaking prescription." -> Do you have an idea why it works? What actually happens inside the network when it is given a symmetry-breaking token? It would be an interesting avenue for a follow-up.

Regarding Fig 6:

12) This figure contains 12 histograms for t t~ + nj, where each of the four rows is for n=1,2,3,4. The last column shows the reconstructed top mass and allows for comparison between different processes. However, each of the other 8 plots shows a different quantity, so one cannot compare the performance of the network for different n. Please explain the selection of plots to show.

13) Why not include uncertainties in this figure?

14) Regarding Fig. 7: When comparing to the paper from which these plots are taken from, the results for the JetGPT architecture have been removed. Why did you decide to do it?

15) Regarding the bibliography: Some of the references, e.g. [33] and [34], seem to be corrupted. There are some names appearing after the titles which do not correspond to the names of the authors of the referenced papers. Please carefully check and correct your bibliography.

16) Regarding Tab. 7: This table contains, amongst other values, the number of parameters of the NN architectures used for the amplitude exercise. One can see that L-GATr has ~26 times more parameters than the MLP model. How can the comparison between these two, and also between the other models, be fair? In other words, how can we know that the difference comes from the actual architecture, not simply from the larger size of the network? Please justify.

17) Regarding Tab. 8: Everywhere else in the study, the Adam optimiser was used, except for the top tagging. Why is that? Please explain.

Typos:

1) page 3, line 3 above Eq. 2: "etric alg" 2) page 5, line 2 below Eq. 9: "representingS" 3) page 6, line 2 from the top: "LayerNorm,tion" 4) page 17, line 4 in the "Performance" paragraph: "appear s"

Reproducibility:

While the focus of this review is the submitted article, it is essential to ensure the reproducibility of the presented results. The authors have made their machine learning model publicly available on GitHub and have clearly invested considerable effort into enhancing its accessibility by providing examples and detailed instructions. This commendable initiative deserves recognition. Nevertheless, the official repository contains certain issues which, although I was ultimately able to resolve, may pose difficulties for less experienced users. Below, I provide a number of suggestions aimed at improving the reproducibility of the code.

  1. There appears to be a bug in the "data/collect_data.py" script on line 71: the order of the arguments passed to the "np.save" function is reversed.

  2. In the second experiment, the configuration specifies "model=gatr_toptagging", but this model does not appear to be available. I assume the intended name is "gatr_tagging".

  3. When attempting to train the top-tagging model on a CPU, I encountered a crash due to the memory-efficient attention mechanism not supporting CPU execution. While it is acceptable for the code to be GPU-only, some exception handling or a clearer error message would be beneficial.

  4. The code expects the file toptagging_mini.npz, but the dataset provided for download did not include this file. I was able to proceed by using toptagging_full.npz instead.

  5. When attempting to run the ttbar generation task using the config_paper configuration, I obtained results only for the 0-jet case. This may be due to an error on my part, but I recommend verifying it.

Recommendation

Ask for major revision

---

## Round 2 · Referee Report · Anonymous (Referee 1) · 2025-5-7

Report

Summary:
The document presents how Lorentz symmetry or Minkowski metric is encoded into the network architecture and its latent representation so it does not have to be learned.

Examples from HEP include regression, classification, and generative tasks and improvement of the performance is demonstrated over the state of the art.

Symmetries are known to be not exact at the measurement level. A learnable symmetry-breaking reference frame is introduced leading to significantly better performance than removing the corresponding equivariance from the network altogether.

Comments:

Chapter 2:
The text is a bit verbose and contains many “for instance”. A lot of examples are presented, and the reader easily misses the red line. A better structured text would be useful to facilitate full appreciation of the work.

The main results are in chapters 3-5.

Chapter 3:

Not clear to the reader what exactly the training inputs are.

Are all encoded symmetries exact? Would be good to state this explicitly.

Fig. 2 is also in Ref 35. The repetition of same results in different publications should be avoided if possible.

Chapter 4:

Table 2 is also in Ref 35. The repetition of same results in different publications should be avoided if possible. The novelty in Table 2 is in the fine-tuned bottom part. Fine-tuning is, however, only detailed in the Appendix. This should be moved to the main body.

Table 3 is difficult to comprehend and deserves more discussion. The metric in the last column almost varies by a factor of two. Not clear to the reader how the default is chosen. Based on the best metric? How will this be done in practice?

Chapter 5:

Also, here what strikes the reader is “The results presented here were briefly discussed in Ref. [35],”
The suggestion is that the abstract and introduction and possibly the title are more explicit about the message of this document wrt Ref 35.

The symmetry is not exact and reference multivector are introduced. The reader wonders if there is an equivalent set of options as presented in Table 3? Or why this is different here.

The reader wonders (as you do yourself) about the conclusion that “enforcing equivariance in the architecture and then allowing the network to break it with reference multivectors outperforms standard non-equivariant networks”
You say yourself that “Strictly speaking, the underlying problem is only symmetric under rotations around the beam axis.”
How is the improvement in Fig. 7 explained? Have you tried a model without the symmetry-breaking multivectors? Such a comparison would make the conclusion more convincing.

Recommendation

Ask for minor revision

---

## Round 2 · Referee Report · Anonymous (Referee 3) · 2025-5-19

Strengths

1- The paper is relevant as it shows the strong performance of Lorentz-Equivariant Geometric Algebra Transformers (L-GATr) in three important contexts of particle physics phenomenology: amplitude regression, jet tagging and Monte Carlo event generators. 2- The paper is well written and clear, and the overlap with Ref. [35] are clearly signalled. 3- The paper demonstrates the superiority of L-GATr performance compared to other available tools, thus paving the way to further applications of these tools to LHC physics.

Weaknesses

1 - There are a few points in which the authors could be more clear and quantitative.
2 - It would be good if the Authors could offer more technical insight on the techniques used to improve the performance.

Report

The authors show that Lorentz-Equivariant Geometric Algebra Transformers (L-GATr) yield a good performance for three important phenomenological applications of Machine Learning to the LHC. The authors first provide a brief mathematical introduction to L-GATr and give some details on how to break the The authors show that Lorentz-Equivariant Geometric Algebra Transformers (L-GATr) yield a good performance for three important phenomenological applications of Machine Learning to LHC physics. The authors first provide a brief mathematical introduction to L-GATr and give some details on how to break the Lorentz symmetry in a L-GATr architecture. Then they show-case the performance of the GAN algorithm in amplitude regression, jet tagging and event generator. The findings presented partially overlap with the ones that the Authors presented in Ref. [35], but for each application they introduce further studies. For example, as far as amplitudes are concerned, they compare the performance of L-GATr against other existing architectures and show that L-GATr achieves the lowest Mean Squared Error for n >=3 jets in the final state. In the case of jet tagging, they show that they can fine-tuned the architecture for top tagging and show the performance improvement associated to pre-training. Finally, for Monte Carlo event generation, they show a very good agreement between the generated data using the L-GATr architecture and events simulated with MadGraph and Pythia.
The paper meets the criteria of relevance, originality and quality required by the Journal and as such I recommend that, after the Authors address the minor points that I raise in my report, the paper is accepted for publication.

Requested changes

1- In the abstract it is not clear that the benchmark is done for Monte Carlo event generators. Same on page 3, when the Authors mention generative network - which can be used in multiple applications - while here the Authors specifically benchmark event generation via generative networks. 2 - Top of page 6, there seems to be a typo “LayerNorm,tion” should probably be “LayerNorm, Attention”. 3 - In Eq. (17) the Authors introduce the normalisation constant \eps, but they do not say what is the value to which they set it and whether it has any effects on the performance of the L-GATr in the applications that they consider. 4 - In the caption of Table 1 they mention the second term in the L-GATr linear layer, but the latter, though it appears explicitly in Eq. (15), is not written in the table. They should either refer to Eq. (15) or write the missing term in the table. 5- In Section 2.4 the Author mention synthetic data. It would be good for them to specify why they use synthetic instead of real data and how the synthetic data are generated. 6- After Eq. (20) the Author mention that they generate 4 10^5 training data points for each multiplicity using MadGraph, they should specify that this is done at LO in QCD. 7- Across the papers the Author estimate the error bands of MSE from the STD of 5 random seeds: is this enough? Would the error change much if they used a bootstrap method to compute it instead of taking the STD over 5 seeds? Also, out of curiosity, why does the error increase so much with the multiplicity for GAP in Fig 2 and not for the other methods? 8 - In Table 2 the authors do not define what the parameter \eps_S correspond to. 9 - In Section 5 the Authors mention that transformer can also be trained jointly on all multiplicities. Would the performance deteriorate much in this case as opposed to the transformed being trained on each multiplicity separately? 10 - When the Authors discuss the relatively low performance around the top pole mass, is there any way forward to improve the performance around that region? 11 - Appendix A:,page 26: why do the validation and testing sets include only 1% of the data? Why later the split becomes 80-10-10 ? 12 - In the table in the appendix, Adam is always used as optimizer, is there a specific reason for this?

Recommendation

Ask for minor revision

---

## Editorial Decision

resubmitted